# A Multi-Constraint Guidance and Maneuvering Penetration Strategy via Meta Deep Reinforcement Learning

**Sibo Zhao** [1], **Jianwen Zhu** [1,2,*], **Weimin Bao** [1,3], **Xiaoping Li** [1] **and Haifeng Sun** [1]

1   School of Aerospace Science and Technology, Xidian University, Xi'an 710126, China;
    zhaosibo@stu.xidian.edu.cn (S.Z.); baoweimin@cashq.ac.cn (W.B.); xpli@xidian.edu.cn (X.L.);
    hfsun@xidian.edu.cn (H.S.)
2   College of Missile Engineering, Rocket Force University of Engineering, Xi'an 710025, China
3   China Aerospace Science and Technology Corporation, Beijing 100048, China
*   Correspondence: jwzhu@xidian.edu.cn; Tel.: +86-18392175968

**Abstract:** In response to the issue of UAV escape guidance, this study proposed a unified intelligent control strategy synthesizing optimal guidance and meta deep reinforcement learning (DRL). Optimal control with minor energy consumption was introduced to meet terminal latitude, longitude, and altitude. Maneuvering escape was realized by adding longitudinal and lateral maneuver overloads. The Maneuver command decision model is calculated based on soft-actor–critic (SAC) networks. Meta-learning was introduced to enhance the autonomous escape capability, which improves the performance of applications in time-varying scenarios not encountered in the training process. In order to obtain training samples at a faster speed, this study used the prediction method to solve reward values, avoiding a large number of numerical integrations. The simulation results demonstrated that the proposed intelligent strategy can achieve highly precise guidance and effective escape.

**Keywords:** gliding flight; UAV penetration; multi-constraint optimal guidance; meta-learning; SAC networks

## 1. Introduction

Hypersonic UAVs mainly glide in near-space [1]. In their early phase, a higher flight velocity is acquired, relying on the thin atmospheric environment, which is an advantage for effectively avoiding the interception of defense systems. At the end of a gliding flight, the velocity is mainly influenced by the aerodynamic force and suffers from the restrictions of heat flow, dynamic pressure, and overload [2]. The velocity advantage leads to penetration becoming more difficult, so orbital maneuvering is applied by UAVs to achieve penetration. The main flight mission is split into avoiding defense system interception and satisfying multiple terminal constraints [3]. The core of this manuscript is designing a penetration guidance strategy via orbital maneuvering capabilities, avoiding interception, and reducing the penetration's impact on guidance accuracy.

The penetration strategy is summarized as a tactical penetration and a technical penetration strategy [4]. The technical penetration strategy changes the flight path through maneuvering, aiming to increase the missed distance in order to successfully penetrate. Common maneuver manners include the sine maneuver, step maneuver, square wave maneuver, and spiral maneuver [5]. There are some limitations and instability for technical penetration strategies, attributed to the UAV struggling to adopt an optimal penetration strategy according to the actual situation of the offensive and defensive confrontations. Compared with the traditional procedural maneuver strategy, the differential game guidance law has the characteristics of real-time operation and intelligence as a tactical penetration strategy [6]. Penetration problems are essentially regarded as the continuous dynamic conflict problem of multi-party participants, and this strategy is an essential solution for

solving multi-party optimal control problems. Applying it to the problem of attack–defense confrontation can not only fully consider the relative information between UAVs and interceptors but also obtain a Nash equation strategy to reduce energy consumption. Many scholars have proposed differential game models of various maneuvering control strategies based on control indexes and motion models. Garcia [7] regarded the scenario of active target defense modeling as a zero-sum differential game, designed a complete differential game solution, and comprehensively considered the optimal strategy of closed-loop state feedback to obtain the value function. In Ref. [8], the optimal guidance problem was studied between an interceptor and an active defense ballistic UAV, and an optimal guidance scheme was proposed based on the linear quadratic differential game method and the numerical solution of Riccati differential equations. Liang [9] mainly analyzed the problem of pursuit and escape attacks of multiple players, inducted the three-body game confrontation into competition and cooperation problems, and solved the optimal solution of multiple players via differential game theory. The above methods are of great significance for analyzing and solving the confrontation process between UAVs and interceptors. Near-space UAVs have the characteristics of high velocity and short time in the phase of attack and defense confrontation terminal guidance [10], and the differential game guidance law struggles to show advantages in this phase. Moreover, the differential game method requires a large amount of calculation, and the bilateral performance indicators are difficult to model [11]; as a result, this theory is unable to be applied in practice.

DRL is a research hotspot in the field of artificial intelligence that has sprung up in recent years, and amazing results have been achieved in robot control, guidance, and control technologies [12]. DRL specifically refers to agents learning in the process of interaction with the environment to find the best strategy to maximize cumulative rewards [13]. With the advantages of dealing with high-dimensional abstract problems and making decisions quickly, DRL provides a new solution for the maneuvering penetration of high-velocity UAVs. In order to solve the problem of intercepting a high maneuvering target, an auxiliary DRL algorithm was proposed in Ref. [14] to optimize the frontal interception guidance control strategy based on a neural network. Simulation results showed that DRL had a higher hit rate and larger terminal interception angle than traditional methods and proximal policy optimization algorithms. Gong [15] proposed an Omni bearing attack guidance law for agile UAVs via DRL, which effectively dealt with aerodynamic uncertainty and strong nonlinearity at a high attack angle. DRL was used to generate a guidance law for the attack angle in an agile turning phase. Furfaro [16] proposed an adaptive guidance algorithm based on classical zero-effort velocity, and the limitations of this algorithm were overcome via RL. A closed-loop guidance algorithm was created that is lightweight and flexible enough to adapt to a given constrained scene.

Compared with differential game theory, DRL is convenient for establishing the performance index function, and it is a feasible method to solve the dynamic programming problem by utilizing the powerful numerical calculation ability of computers to skillfully avoid solving the function analytical solution. However, the traditional DRL has some limitations, such as high sample complexity, low sample utilization, a long training time, and so on. Once the mission changes, the original DRL parameters are hard to adapt to the new mission and need to be learned from scratch. A change in mission or environment will lead to the failure of the trained model and poor generalization ability of the model. In order to solve the existing problems in DRL, researchers introduced meta-learning into DRL and proposed Meta DRL [17]. Lu et al. [18] mainly solved the issues of maximizing the total data collected and avoiding collisions during the guidance flight and improved the adaptability to different missions via meta RL. Hu et al. [19] mainly studied a challenging trajectory design problem, optimized the DBS trajectory, and considered the uncertainty and dynamics of terrestrial users' service requests. MAML was introduced for the purposes of solving raised POMDPS and enhancing adaptability to balance flight aggressiveness and safety [20]. For suspended payload transportation tasks, the paper [21] proposed a meta-

learning approach to improve adaptability. The simulation demonstrated improvements in closed-loop performance compared to non-adaptive methods.

By learning useful meta knowledge from a group of related missions, agents acquire the ability to learn, and the learning efficiency on new missions is improved and the complexity of samples is reduced. When faced with new missions or environments, the network responds quickly based on the previously accumulated knowledge, so only a small number of samples are needed to quickly adapt to the new mission.

Based on the above analysis, this manuscript proposes Meta DRL to solve the UAV guidance penetration strategy, and the DRL is improved, resulting in enhancing the adaptability of UAVs in complex and changeable attack and defense confrontations. In addition, the idea of meta-learning is used to enable UAVs to learn and improve their ability for autonomous flight penetration. The core contributions of this manuscript are as follows:

1. By modeling the three-dimensional attack and defense scene between UAVs and interceptors, we analyze the terminal and process constraints of UAVs. A guidance penetration strategy based on DRL is proposed, aiming to provide the optimal solution for maneuvering penetration under a constant environment or mission.
2. Meta-learning is used to improve the UAV guidance penetration strategy. The improvement enables the UAV to learn and enhances the autonomous penetration ability.
3. Through the simulation analysis, the manuscript analyzes the penetration strategy, explores penetration timing and maneuvering overload, and summarizes the penetration tactics.

## 2. Modeling of the Penetration Guidance Problem

### 2.1. Modeling of UAV Motion

The three-degree-of-freedom motion equation is adopted to describe UAVs and the dynamic equation is established in the ballistic coordinating system:

$$
\begin{cases}
\dot{v} = -\frac{\rho v^2 S_m C_D}{2m} + g'_r \sin\theta + g_{\omega e}(\cos\sigma\cos\theta\cos\phi + \sin\theta\sin\phi) \\
\quad + \omega_e^2 r(\cos^2\phi\sin\theta - \cos\phi\sin\phi\cos\sigma\cos\theta) \\
\dot{\theta} = \frac{\rho v^2 S_m C_L \cos v}{2mv} + \frac{g'_r \cos\theta}{v} - 2\omega_e\sin\sigma\cos\phi + \frac{v\cos\theta}{r} \\
\quad + \frac{g_{\omega e}}{v}(\cos\theta\sin\phi - \cos\sigma\sin\theta\cos\phi) + \frac{\omega_e^2 r}{v}(\cos\phi\sin\phi\cos\sigma\sin\theta + \cos^2\phi\cos\theta) \\
\dot{\sigma} = -\frac{\rho v^2 S_m C_L \sin v}{2mv\cos\theta} - \frac{g_{\omega e}\sin\sigma\cos\phi}{v\cos\theta} + \frac{\omega_e^2 r(\cos\phi\sin\phi\sin\sigma)}{v\cos\theta} + \frac{v\tan\phi\cos\theta\sin\sigma}{r} \\
\quad - 2\omega_e(\sin\phi - \cos\sigma\tan\theta\cos\phi) \\
\dot{\phi} = \frac{v\cos\theta\cos\sigma}{r} \\
\dot{\lambda} = -\frac{v\cos\theta\sin\sigma}{r\cos\phi} \\
\dot{r} = v\sin\theta
\end{cases}
\tag{1}
$$

where $v$ is the velocity of UAV relative to the earth, $\theta$ is the velocity slope angle, $\sigma$ is the velocity azimuth, and the positive direction is clockwise from the north. $r$ represents the geocentric distance, and $(\lambda, \phi)$ is the longitude and latitude. The differential argument is flight time $t$. $g_{\omega e}$ is the component of the Earth's gravitational acceleration in the direction of the Earth's rotational angular rate $\omega_e$, while $g'_r$ is the component of the Earth's gravitational acceleration in the direction of the geo-center. $\rho$ is the density of the atmosphere, while $m$ and $S_m$ are the mass and reference area of UAV. $C_D$ and $C_L$ are the drag and lift coefficient, respectively, relating to the Mach number and attack angle, so the control variable attack angle $\alpha$ is implicit in it, and the other control variable is the bank angle $v$.

For a hypersonic UAV with a large lift–drag ratio (*L/D*), the heat flow, overload, and dynamic pressure are considered as flight process constraints, as follows:

$$
\begin{cases}
k_h \rho^{1/2} v^3 \leq Q_{smax} \\
\frac{\rho v^2}{2} \leq q_{max} \\
\frac{\sqrt{D^2 + L^2}}{mg_0} \leq n_{max}
\end{cases}
\tag{2}
$$

where $Q_{smax}$, $q_{max}$, and $n_{max}$ are the maximum heat flow, dynamic pressure, and overload, respectively; the parameter $k_h$ is a constant coefficient, and $g_0$ is the gravitational acceleration at sea level. In order to keep the gliding state steady and effectively prevent trajectory jumps, the balance of lift and gravity is required by UAV [22]. Stable gliding is generally considered as quasi-equilibrium gliding (QEGs). At present, the international definition of QEG can be summarized into two types [23]: the velocity slope angle is considered as constant, expressed by $\dot{\theta} = 0$, or the flight altitude variance ratio is considered as constant, expressed by $\ddot{h} = 0$. Moreover, $\dot{\theta} = 0$ was adopted by traditional QEG guidance, and $\ddot{h} = 0$ mainly appeared in the analytical prediction guidance of the Mars landing at the end of the last century [24].

For high *L/D* UAVs flying in the gliding phase, the Coriolis inertial force and centrifugal inertial force caused by the Earth's rotation are relatively small compared to aerodynamic forces. Therefore, in the design of guidance laws, the Earth can be assumed to be a homogeneous sphere that does not rotate. We further decompose Equation (1) into longitudinal and lateral directions, where the longitudinal motion equation is as follows:

$$\begin{cases} \dot{v} = -\frac{\rho v^2 S_m C_D}{2m} - g\sin\theta \\ \dot{\theta} = \frac{\rho v^2 S_m C_L \cos v}{2mv} - \frac{g\cos\theta}{v} + \frac{v\cos\theta}{r} \\ \dot{h} = v\sin\theta \\ \dot{L}_R = R_e v\cos\theta/(R_e + h) \end{cases} \tag{3}$$

where $h = r - R_e$ represents the flight altitude, and $L_R$ is range. For a gliding UAV flying in close proximity to space, $h \ll R_e$. Therefore, range differentiation can be further described as follows:

$$\dot{L}_R = v\cos\theta \tag{4}$$

This manuscript adopts $\dot{\theta} = 0$ as the QEG condition. Referring to the second equation in Equation (3), $\dot{\theta} = 0$ is transferred into Equation (5).

$$\begin{aligned} \dot{\theta} = 0 &= \frac{\rho v^2 S_m C_L \cos v}{2mv} - \frac{g\cos\theta}{v} + \frac{v\cos\theta}{r} \\ &\Rightarrow \frac{\rho v^2 S_m C_L \cos v}{2mv} = \frac{g\cos\theta}{v} - \frac{v\cos\theta}{r} \\ &\Rightarrow \frac{\rho v^3 S_m C_L \cos v}{2} = \frac{mg\cos\theta}{v} - \frac{mv^2\cos\theta}{r} \end{aligned} \tag{5}$$

where $L = \frac{\rho v^3 S_m C_D}{2}$, hence

$$m\left(g - \frac{v^2}{r}\right)\cos\theta - L\cos v = 0 \tag{6}$$

### 2.2. Description of Flight Missions

2.2.1. Guidance Mission

The physical meaning of gliding guidance is eliminating heading errors, satisfying complex process constraints, and minimizing energy loss. The UAV is guided to glide unpowered to the setting terminal target point $(h_f, \lambda_f, \phi_f)$, satisfying the terminal altitude, longitude and latitude. Hence, the terminal constraints are expressed by Equation (7).

$$\begin{cases} h(L_{Rf}) = h_f, \ \lambda(L_{Rf}) = \lambda_f \\ \phi(L_{Rf}) = \phi_f, \Delta\sigma_f \leq \Delta\sigma_{max} \end{cases} \tag{7}$$

where the terminal range $L_{Rf}$ is given, $\Delta\sigma$ represents the heading error, and $\Delta\sigma_{max}$ is a pre-setting allowable value. The guidance problem is the process of determining $\alpha$ and $v$.

2.2.2. Penetration Mission

The main indexes are used to judge penetration probability as follows:

(1) The miss distance $D_{miss}$ with the interceptor at the encounter moment.

(2) The overload $N_m$ of the interceptor in the last phase.

(3) The line-of-sight (LOS) angular rates $\dot{\theta}_{intlos}$ and $\dot{\sigma}_{intlos}$ with the interceptor at the encounter moment.

where $D_{miss}$ directly reflects the result of penetration—the larger the $D_{miss}$, the greater of penetration probability. The $N_m$ and LOS angle indirectly reflect the result of penetration. The larger the $\dot{\theta}_{intlos}$ and $\dot{\sigma}_{intlos}$, the more difficult it is for the interceptor to successfully intercept, which is attributed to $\theta_{intlos}$ and $\sigma_{intlos}$ not easily converging to a constant value at the encounter moment. In addition, a larger overload is required to adjust $\theta_{intlos}$ and $\sigma_{intlos}$ at the end of interception. The larger the $N_m$, the greater the control cost that the interceptor has to pay to complete the interception mission. Once $N_m$ exceeds the overload limit of the interceptor, it indicates that the control is saturated, which demonstrates that the interceptor struggles to intercept based on the current maximum overload constraint. Referring to the size and flight characteristics of UAVs [25], this manuscript assumes that the penetration mission is completed if $D_{miss}$ is greater than 2 m.

The maximum maneuvering overload of UAVs is constrained by the structure and QEG condition [24]. If the lateral maneuver overload is too large, it will cause the UAV to deviate from the course, eventually leading to the failure of the guidance mission. A large longitudinal maneuver amplitude will have a significant impact on the L/D of the UAV, affecting safety. According to the above analysis, the maximum lateral maneuvering overload is set as 2 g, and the maximum longitudinal maneuvering overload is set as 1 g.

*2.3. The Guidance Law of Interceptor*

The guidance law of the interceptor relies on the inertial navigation system to obtain information such as the position and velocity of the UAV. The relative motion is shown in Figure 1.

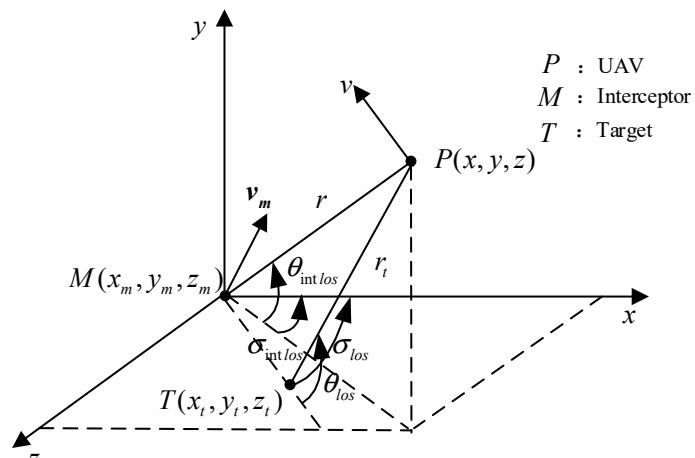

**Figure 1.** Attack–defense model.

P, T, and M, respectively, represent the UAV, target, and interceptor. $r$ is the relative position between the UAV and the interceptor, and $r_t$ is the target.

The LOS angular rate and the approach velocity to the UAV are obtained for the interceptor. Overload control command is derived via generalized proportional navigation guidance (GPNG).

$$\begin{cases} n^*_{y_2} = K_D |\dot{r}| \dot{\theta}_{intlos} / g_0 \\ n^*_{z_2} = K_T |\dot{r}| \dot{\sigma}_{intlos} / g_0 \end{cases} \tag{8}$$

As shown in Equation (8), $K_D$ and $K_T$ are navigation ratios in the longitudinal and lateral direction, respectively. $\dot{r}$ is the approach velocity. $\dot{\theta}_{\text{int}los}$ and $\dot{\sigma}_{\text{int}los}$ represent the LOS angular rate in the longitudinal and lateral directions, respectively.

$$\begin{cases} r = \sqrt{(x-x_m)^2 + (y-y_m)^2 + (z-z_m)^2} \\ \theta_{\text{int}los} = \arcsin((y-y_m)/r) \\ \sigma_{\text{int}los} = \arctan((z-z_m)/(x-x_m)) \end{cases} \tag{9}$$

Differentiating Equation (9) with respect to $t$, we obtain the following:

$$\begin{cases} \dot{\theta}_{\text{int}los} = \frac{1}{\sqrt{1-(\frac{y-y_m}{r})^2}} \cdot \frac{(\dot{y}-\dot{y}_m)r-(y-y_m)\dot{r}}{r^2} \\ \dot{\sigma}_{\text{int}los} = \frac{1}{1+(\frac{z-z_m}{x-x_m})^2} \cdot \frac{(\dot{z}-\dot{z}_m)\cdot(x-x_m)-(z-z_m)\cdot(\dot{x}-\dot{x}_m)}{(x-x_m)^2} \\ \dot{r} = \frac{(x-x_m)\cdot(\dot{x}-\dot{x}_m)+(y-y_m)\cdot(\dot{y}-\dot{y}_m)+(z-z_m)\cdot(\dot{z}-\dot{z}_m)}{r} \end{cases} \tag{10}$$

## 3. Design of Penetration Strategy Considering Guidance

### 3.1. Guidance Penetration Strategy Analysis

Generally, the UAV achieves penetration through a velocity advantage or increased maneuvering overload. The former is used in pre-gliding flight, and compared with interceptors, the velocity of UAVs is relatively large, which is conducive to penetrating defenses. More threats to UAVs come from the defense systems of intended targets, resulting in an intercept threat focusing on the end of the glide flight. However, the flight velocity gradually decreases, which is not enough to penetrate escape. Based on the above analysis, the manuscript designs a penetration strategy by increasing the maneuvering overload.

Firstly, in previous research [26], the energy-optimized gliding guidance law was designed. In this study, avoiding the intercept by maneuvering becomes the key point. The real-time flight information of the interceptor cannot easily be accurately obtained by the UAV, while the real-time flight information of the UAV is obtained by the interceptor. Based on the UAV penetration mission, which only knew the initial launch position of the interceptor, this manuscript simulates the attack and defense environment between the UAV and the interceptor, applying the DRL method to solve the overload command of UAV maneuvering penetration. DNN parameters are trained offline, and a maneuvering penetration command is constructed online. The launch position of the interceptor is changeable, and a stable penetration command cannot adapt to the complex penetration environment. To improve the adaptability of DNN parameters, the original DRL is optimized by adopting the idea of meta-learning, and the ontology and environmental information are fully utilized. The optimization of meta-learning enhances flight capability, fast response to complex missions, and flight self-learning. Finally, the UAV guidance penetration strategy based on Meta DRL is proposed in this manuscript, as shown in Figure 2.

As for the gliding guidance mission, DRL and optimal control are used to achieve the guidance penetration mission of the UAV. An optimal guidance command was conducted in previous research [26], which was introduced to satisfy constraints regarding terminal position, altitude, and minimal energy loss. Maneuvering overloads are added between the longitudinal and lateral directions, aiming to achieve the penetration mission at the end of the gliding flight. Maneuvering overloads are solved via DRL. The DNN parameters are optimized by means of meta-learning.

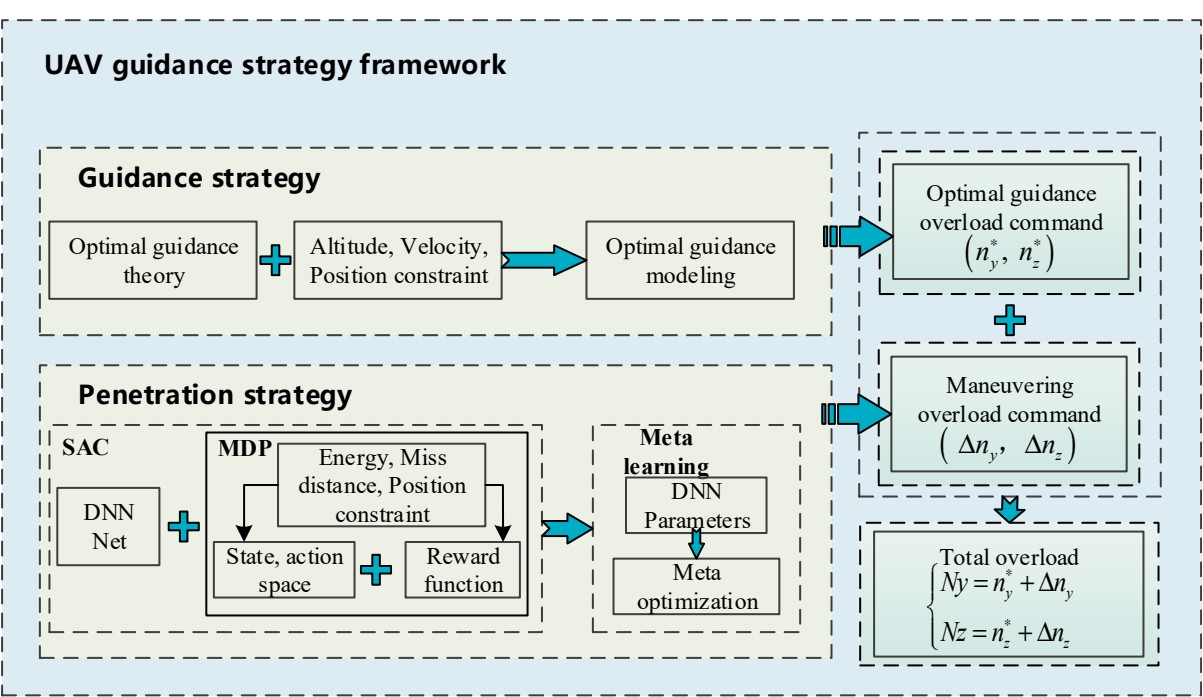

**Figure 2.** Guidance penetration strategy.

Guidance command $\left(n_y^*,\ n_z^*\right)$ is generated by the optimal guidance strategy, and maneuvering overload command $\left(\Delta n_y, \Delta n_z\right)$ is generated by meta DRL. The flight total overload is shown in Equation (11).

$$\begin{cases} Ny = n_y^* + \Delta n_y \\ Nz = n_z^* + \Delta n_z \end{cases} \tag{11}$$

Maneuvering penetration command via meta DRL is the core of this manuscript. The penetration considering guidance is described as a Markov decision process (MDP), which consists of finite-dimensional continuous flight state space, longitudinal and lateral direction overload sets, and a reward function judging the penetration strategy. Flight data generated via numerical integration and SAC networks are introduced to train and learn MDP. Optimizing network parameters via meta-learning aims at adjusting network parameters with very little flight data when the UAV is faced with online mission changes, adapting to the new environment as soon as possible.

*3.2. Energy Optimal Gliding Guidance Method*

In previous research [26], based on the QEG condition and taking the required overload as the control amount, the performance index with minimum energy loss was established. The optimal longitudinal and lateral overload were designed, respectively, satisfying the constraints on terminal latitude, longitude, altitude, and velocity. The required overload command is shown in Equation (12).

$$\begin{cases} u_y^* = k(C_h L_R - C_\theta) + 1 \\ u_z^* = \dfrac{\sigma_{los} - \sigma}{k\left(L_{Rf} - L_R\right)} \end{cases} \tag{12}$$

where $u_y^* = n_y^*$ and $u_z^* = n_z^*$ are the optimal overload between the longitudinal and lateral dimensions. $k = \frac{g_0}{v^2} \approx \frac{g}{v^2}$, $L_R$ is the current range, and $L_{Rf}$ is the total range of the gliding

phase. $C_h$ and $C_\theta$ are the guidance coefficients based on optimal control, represented as Equation (13).

$$\begin{cases} C_h = \dfrac{6\left(\left(L_R - L_{Rf}\right)\left(\theta_f + \theta\right) - 2h + 2h_f\right)}{k^2\left(L_R - L_{Rf}\right)^3} \\ C_\theta = \dfrac{2\left(L_R L_{Rf}\left(\theta - \theta_f\right) - L_{Rf}^2\left(2\theta + \theta_f\right) + L_R^2\left(2\theta_f + \theta\right) + 3\left(L_{Rf} + L_R\right)\left(h_f - h\right)\right)}{k^2\left(L_R - L_{Rf}\right)^3} \end{cases} \tag{13}$$

Based on Equation (14), the control variables $\alpha$ and $v$ are calculated as follows:

$$\begin{cases} \dfrac{\rho v^2 S_m C_L(Ma,\alpha)}{2g_0} = \sqrt{n_y^{*2} + n_z^{*2}} \\ v = \arctan\left(\dfrac{n_z^*}{n_y^*}\right) \end{cases} \tag{14}$$

where $\alpha$ is obtained by calculating the contrast value in Equation (14).

## 4. RL Model for Penetration Guidance

The problem of maneuvering penetration is modeled as a series of stationary MDPs [27] with unknown transition probabilities. The continuous flight state space, action set, and reward function for judging the command are determined in this section.

### 4.1. MDP of the Penetration Guidance Mission

Deterministic MDP with a continuous state and action is defined as $(S, A, T, R, \gamma)$ via a quintuple. $S$ is described as the continuous state space, $A$ is described as the finite action set, and $T$ is depicted as the state transition function. $S \times A \rightarrow T$, reflecting deterministic state transition relationships. $R$ is defined as the immediate reward. $\gamma \in [0, 1]$ is the discount factor, to balance immediate and forward reward.

The agent chooses the action $a_t \in A$ at the current state $s_t$, while the state changes from $s_t$ to $s_{t+1} \in S$, and the environment returns an immediate reward $R_t = f(s_t, a_t)$ for the agent. Cumulative rewards are obtained with a controlled action sequence $\tau$, as shown in Equation (15).

$$G(s_0, \tau) = R_0 + \gamma R_1 + \gamma^2 R_2 + \dots = \sum_{t=0}^{\infty} \gamma^t R^t \tag{15}$$

The goal of MDP is determining policy, and maximizing the expected accumulated rewards, as follows:

$$\tau^* = \arg\max_\tau \{ G(s_0, \tau) \} \tag{16}$$

#### 4.1.1. State Space Design

The environment of attack–defense is abstracted as a state space of MDP, which applies guidance for action commands. UAVs struggle to acquire the flight information and guidance laws of interceptors. Hence, the information of the UAV and the target is only considered as state space.

$$S = \left\{ x_r, y_r, z_r, \theta_{los}, \sigma_{los} \right\} \tag{17}$$

where $(x_r, y_r, z_r)$ represents the relative distance between the UAV and the target under the North East Down (NED) Coordinate System. $(\theta_{los}, \sigma_{los})$ represents, respectively, the longitudinal and lateral LOS angles. In order to eliminate dimension differences and enhance compatibility among states, $S$ is normalized using Equation (18).

$$S = \left( x_r = \frac{x_r}{x_{r0}},\ y_r = \frac{y_r}{y_{r0}},\ z_r = \frac{z_r}{z_{r0}},\ \theta_{los} = \frac{\theta_{los}}{2\pi},\ \sigma_{los} = \frac{\sigma_{los}}{2\pi} \right) \tag{18}$$

### 4.1.2. Action Space Design

Action is a decision selected by the UAV based on the current state, and action space $A$ is the set of all possible decisions. Overload directly affects the velocity azimuth and slope angle and indirectly affects the gliding flight status; hence, this manuscript determines overload as an intermediate control variable.

On the basis of the optimal guidance law and flight process constraints, longitudinal overload $\Delta n_y$ and lateral overload $\Delta n_z$ are added to the action space of MDP. Considering the safety of the UAV and heading error, this manuscript assumes that the longitudinal and lateral maximum maneuvering overloads are 1 g and 2 g, respectively. $A$ is defined as a continuous set, shown in Equation (19).

$$A = \left\{ \begin{array}{l} \Delta n_y \in [-g,\ g] \\ \Delta n_z \in [-2g,\ 2g] \end{array} \right. \tag{19}$$

### 4.2. Multi-Missions Reward Function Designing

The reward function $f_r$ is an essential section for guiding and training the maneuvering penetration strategy. After execution of the action command, $f_r$ returns a reward value to the UAV, which reflects the fairness and rationality of the action judgment. The rationality of $f_r$ directly affects the training result and determines the efficiency of SAC training. In this manuscript, the aim of $f_r$ is to guide the UAV to achieve the guidance penetration mission while satisfying terminal multi-constraints. Given the requirements of the mission, $f_r$ consists of the miss distance with the interceptor and the terminal deviation from the target.

$$f_r(s, a) = c_1 d_{miss} - c_2 d_{error} \tag{20}$$

where $d_{miss}$ and $d_{error}$ are the miss distance and terminal deviation after execution the action command. The sufficient and necessary condition of satisfying terminal position deviation is eliminating the heading error, which is directly related to the LOS angular rate. Similarly, $d_{miss}$ can be reflected by the LOS angular rate at the encounter time between the UAV and the interceptor. Therefore, the normalized $f_r$ is expressed by Equation (21).

$$f_r = c_1 \sqrt{\left(\overline{\dot{\theta}}_{intlos}\right)^2 + \left(\overline{\dot{\sigma}}_{intlos}\right)^2} - c_2 \sqrt{\left(\overline{\dot{\theta}}_{los}\right)^2 + \left(\overline{\dot{\sigma}}_{los}\right)^2} \tag{21}$$

where $\overline{\dot{\theta}}_{intlos}$ represents the normalized LOS angular rate in the longitudinal direction with the interceptor, $\overline{\dot{\sigma}}_{intlos}$ represents the normalized LOS angular rate in the lateral direction with the interceptor. $\overline{\dot{\theta}}_{los}$ represents the normalized LOS angular rate in the longitudinal direction with the target, and $\overline{\dot{\sigma}}_{los}$ represents the normalized LOS angular rate in the lateral direction with the target. In this manuscript, the LOS angular rates at the encounter time and terminal time are solved analytically via numerical calculation.

### 4.2.1. The Solution of LOS Angular Rate in the Lateral Direction

The attack–defense confrontation model in the lateral direction is shown in Figure 3.

P, T, and M, respectively represent UAV, target, and interceptor. $L_{Rgo}$ and $L_{Mgo}$ are represented as the remaining range among UAV, target and interceptor, which are calculated in Equations (22) and (23).

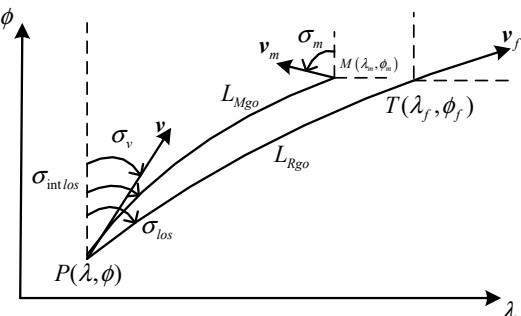

**Figure 3.** The attack–defense confrontation model in the lateral direction.

The lateral relative motion model between UAV and the target is shown in Equation (22).

$$\begin{cases} \dot{L}_{Rgo} = -v \cos \Delta \delta \\ L_{Rgo} \dot{\sigma}_{los} = v \sin \Delta \delta \end{cases} \tag{22}$$

where $\Delta \delta = \sigma - \sigma_{los}$, represents the heading error between UAV and the target point. Lateral relative motion model between UAV and interceptor is shown in Equation (23).

$$\begin{cases} \dot{L}_{Mgo} = -v \cos \Delta \sigma - v_m \cos \Delta \sigma_m \\ L_{Mgo} \dot{\sigma}_{intlos} = v \sin \Delta \sigma - v_m \sin \Delta \sigma_m \end{cases} \tag{23}$$

where $\Delta \sigma = \sigma - \sigma_{intlos}$, represents heading error between UAV and the interceptor. In order to simplify the calculation, the relative motion equation Equation (23) is conducted as Equation (24):

$$\begin{cases} \dot{L}_{Mgo} = -v_r \cos \Delta \sigma_{mn} \\ L_{Mgo} \dot{\sigma}_{intlos} = v_r \sin \Delta \sigma_{mn} \end{cases} \tag{24}$$

where $v_r$ and $\Delta \sigma_{mn}$ are calculated as Equation (25).

$$\begin{cases} v_r = \sqrt{\left(v \cos \sigma_v - v_m \cos \sigma_m\right)^2 + \left(v \sin \sigma_v - v_m \sin \sigma_m\right)^2} \\ \Delta \sigma_{mn} = \mathrm{atan} \frac{v \sin \sigma_v - v_m \sin \sigma_m}{v \cos \sigma_v - v_m \cos \sigma_m} \end{cases} \tag{25}$$

To facilitate the analysis and prediction, $\dot{\sigma}_{los}$ is calculated as follows, taking the derivation of the second formula in Equation (22):

$$\dot{L}_{Rgo} \dot{\sigma}_{LOS} + L_{Rgo} \ddot{\sigma}_{LOS} = \dot{v} \sin \Delta \sigma + v \Delta \dot{\sigma} \cos \Delta \sigma \tag{26}$$

Bring the heading error and first formula in Equation (22) into Equation (26), the rate of LOS angular rate is calculated by Equation (27).

$$\begin{aligned} \dot{L}_{Rgo} \dot{\sigma}_{los} + L_{Rgo} \ddot{\sigma}_{los} &= \dot{v} \sin \Delta \sigma + v \Delta \dot{\sigma} \cos \Delta \sigma \Rightarrow \\ \dot{L}_{Rgo} \dot{\sigma}_{los} + L_{Rgo} \ddot{\sigma}_{los} &= \frac{\dot{v}}{v} L_{Rgo} \dot{\sigma}_{los} - \dot{L}_{Rgo} \dot{\sigma}_{los} + \dot{L}_{Rgo} \dot{\sigma} \Rightarrow \\ \ddot{\sigma}_{los} &= \left( \frac{\dot{v}}{v} - \frac{2 \dot{L}_{Rgo}}{L_{Rgo}} \right) \dot{\sigma}_{los} + \frac{\dot{L}_{Rgo}}{L_{Rgo}} \dot{\sigma} \end{aligned} \tag{27}$$

Defining $T_{goc}$ as the predicted remaining time of flight, $T_{goc}$ is derived via the remaining flight range and variation in range, expressed in Equation (28).

$$T_{goc} = -\frac{L_{Rgo}}{\dot{L}_{Rgo}} \tag{28}$$

Defining the value of state $x = \dot{\sigma}_{los}$ and the value of control $u = \dot{\sigma}$, the differential of LOS angular rate is obtained, which is shown in Equation (29).

$$\dot{x} = \left(\frac{\dot{v}}{v} + \frac{2}{T_{goc}}\right)x - \frac{1}{T_{goc}}u \tag{29}$$

In the latter phase of glide flight, $\frac{\dot{v}}{v}$ is an order of magnitude smaller than $\frac{2}{T_{goc}}$. Equation (29) is further simplified to Equation (30).

$$\dot{x} = \frac{2}{T_{goc}}x - \frac{1}{T_{goc}}u \tag{30}$$

The current remaining flight time $T_{goc}$, as shown in Equation (31). a certain time $t$ of future flight starting from the current time $t$, and the remaining flight time $T_{go1}$ at time $t$ satisfy the following relationship:

$$T_{goc} = T_{go1} + t \tag{31}$$

$dT_{go1} = -dt$ represents the derivation of remaining flight time. For a given control input $u$, the definite integral of Equation (30) is solved, and the calculated result is shown in Equation (32).

$$
\begin{aligned}
x(t) &= e^{\int \frac{2}{T_{go1}}dt}\left(\int -\frac{u}{T_{go1}}e^{-\int \frac{2}{T_{go1}}dt}dt + C\right) \\
&= e^{\int \frac{2}{T_{goc}-t}dt}\left(\int -\frac{u}{T_{goc}-t}e^{-\int \frac{2}{T_{goc}-t}dt}dt + C\right) \\
&= e^{-2\ln(T_{goc}-t)}\left(\int \frac{u}{t-T_{goc}}e^{2\ln(t-T_{goc})}dt + C\right) \\
&= \frac{1}{(T_{goc}-t)^2}\left(\int u(t-T_{goc})dt + C\right) \\
&= \frac{1}{(T_{goc}-t)^2}\left(u\left(\frac{1}{2}t^2 - T_{goc}t\right) + C\right)
\end{aligned}
\tag{32}
$$

The LOS angular rate is $\dot{\sigma}_{los}$, at the current time $t = 0$, the constant $C$ is expressed by Equation (33).

$$C = \dot{\sigma}_{los}T_{goc}^2 \tag{33}$$

$\dot{\sigma}_{los}$ is obtained by Equation (34).

$$\dot{\sigma}_{los}(t) = \frac{1}{(T_{goc}-t)^2}\left(u\left(\frac{1}{2}t^2 - T_{goc}t\right) + \dot{\sigma}_{los}T_{goc}^2\right) \tag{34}$$

$\dot{\sigma}_{los}$ at the terminal time $t_f$ is shown in Equation (35).

$$\dot{\sigma}_{los} = \dot{\sigma}_{los}\left(t_f\right) = \frac{1}{\left(T_{goc}-t_f\right)^2}\left(u\left(\frac{1}{2}t_f^2 - T_{goc}t_f\right) + \dot{\sigma}_{los}T_{goc}^2\right) \tag{35}$$

Similarly, based on above analysis, $\dot{\sigma}_{intlos}$ is calculated via the analysis and prediction. The solution is shown in Equation (36).

$$\dot{\sigma}_{intlos} = \dot{\sigma}_{intlos}\left(t_{intf}\right) = \frac{1}{\left(T_{intgoc}-t_{intf}\right)^2}\left(u\left(\frac{1}{2}t_{intf}^2 - T_{intgoc}t_{intf}\right) + \dot{\sigma}_{intlos}T_{intgoc}^2\right) \tag{36}$$

where $T_{intgoc}$ represents the total encounter time based on the current moment, $t_{intf}$ represents encounter time with interceptor, and $u$ is the input overload.

4.2.2. The Solution of LOS Angular Rate in the Longitudinal Direction

The attack–defense confrontation model in longitudinal direction is shown in Figure 4.

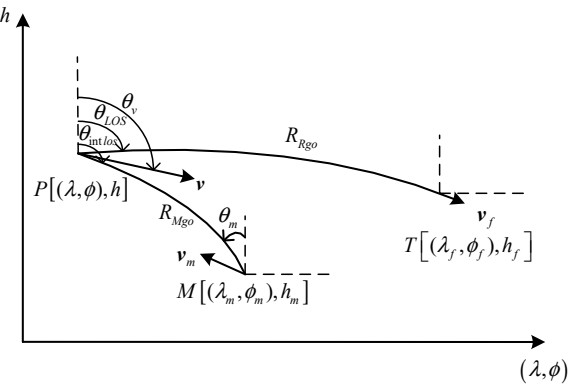

**Figure 4.** The attack–defense confrontation model in longitudinal direction.

$R_{Rgo}$ and $R_{Mgo}$ represent the remaining range among the UAV, target and interceptor, which are calculated in Equations (37) and (38).

The longitudinal relative motion model between the UAV and the target is shown in Equation (22).

$$\begin{cases} \dot{R}_{Rgo} = -v \cos \Delta\theta \\ R_{Rgo}\dot{\theta}_{los} = v \sin \Delta\theta \end{cases} \tag{37}$$

The longitudinal relative motion model between the UAV and the interceptor is shown in Equation (38).

$$\begin{cases} \dot{R}_{Mgo} = -v \cos \Delta\theta - v_m \cos \Delta\theta_m \\ \dot{R}_{Mgo}\dot{\theta}_{intlos} = v \sin \Delta\theta - v_m \sin \Delta\theta_m \end{cases} \tag{38}$$

In order to simplify the calculation, the relative motion equation Equation (38) is expressed as Equation (39).

$$\begin{cases} \dot{R}_{Mgo} = -v_r \cos \Delta\theta_{mn} \\ R_{Mgo}\dot{\theta}_{intlos} = v_r \sin \Delta\theta_{mn} \end{cases} \tag{39}$$

where $v_r$ and $\Delta\theta_{mn}$ are calculated in Equation (40).

$$\begin{cases} v_r = \sqrt{(v \cos \theta_v - v_m \cos \theta_m)^2 + (v \sin \theta_v - v_m \sin \theta_m)^2} \\ \Delta\theta_{mn} = \mathrm{atan} \frac{v \sin \theta_v - v_m \sin \theta_m}{v \cos \theta_v - v_m \cos \theta_m} \end{cases} \tag{40}$$

To facilitate the analysis and prediction, $\dot{\theta}_{los}$ is calculated as follows. Taking the derivation of the second formula in Equation (37) results in Equation (41):

$$\dot{R}_{Rgo}\dot{\theta}_{LOS} + R_{Rgo}\ddot{\theta}_{LOS} = \dot{v} \sin \Delta\theta + v\Delta\dot{\theta} \cos \Delta\theta \tag{41}$$

Bring the heading error and first formula in Equation (37) results in Equation (41), the rate of the LOS angular rate is calculated using Equation (42).

$$\begin{aligned} \dot{R}_{Rgo}\dot{\theta}_{los} + R_{Rgo}\ddot{\theta}_{los} &= \dot{v} \sin \Delta\theta + v\Delta\dot{\theta} \cos \Delta\theta \Rightarrow \\ \dot{R}_{Rgo}\dot{\theta}_{los} + R_{Rgo}\ddot{\theta}_{los} &= \frac{\dot{v}}{v}R_{Rgo}\dot{\theta}_{los} - \dot{R}_{Rgo}\dot{\theta}_{los} + \dot{R}_{Rgo}\dot{\theta} \Rightarrow \\ \ddot{\theta}_{los} &= \left(\frac{\dot{v}}{v} - \frac{2\dot{R}_{Rgo}}{R_{Rgo}}\right)\dot{\theta}_{los} + \frac{\dot{R}_{Rgo}}{R_{Rgo}}\dot{\theta} \end{aligned} \tag{42}$$

Based on the predicted remaining time of flight $T_{goc}$ in lateral prediction, the LOS angular rate is $\dot{\sigma}_{los}$, and at the current time $t = 0$, the constant $C$ is expressed by Equation (43).

$$C = \dot{\theta}_{los} T_{goc}^2 \tag{43}$$

$\dot{\theta}_{los}$ is obtained using Equation (44).

$$\dot{\theta}_{los}(t) = \frac{1}{\left(T_{goc} - t\right)^2} \left( u\left(\frac{1}{2}t^2 - T_{goc}t\right) + \dot{\theta}_{los} T_{goc}^2 \right) \tag{44}$$

$\dot{\theta}_{los}$ at the terminal time $t_f$ is shown in Equation (45).

$$\dot{\theta}_{los} = \dot{\theta}_{los}\left(t_f\right) = \frac{1}{\left(T_{goc} - t_f\right)^2} \left( u\left(\frac{1}{2}t_f^2 - T_{goc}t_f\right) + \dot{\theta}_{los} T_{goc}^2 \right) \tag{45}$$

Similarly, based on the above analysis, $\dot{\theta}_{intlos}$ is calculated via the analysis and prediction. The solution is shown in Equation (46).

$$\dot{\theta}_{intlos} = \dot{\theta}_{intlos}\left(t_{intf}\right) = \frac{1}{\left(T_{intgoc} - t_{intf}\right)^2} \left( u\left(\frac{1}{2}t_{intf}^2 - T_{intgoc}t_{intf}\right) + \dot{\theta}_{intlos} T_{intgoc}^2 \right) \tag{46}$$

where $T_{intgoc}$ represents the total encounter time based on the current moment, $t_{intf}$ represents encounter time with interceptor, and $u$ is the input overload.

According to the above analysis, $\left(\dot{\sigma}_{los}, \dot{\theta}_{los}\right)$ and $\left(\dot{\sigma}_{intlos}, \dot{\theta}_{intlos}\right)$ depend on the change in the LOS angle rate. For the guidance mission, $\left(\dot{\sigma}_{los}, \dot{\theta}_{los}\right)$ is related to the overload of the UAV, and the smaller the value, the closer the UAV approaches the target at the end of the gliding flight. For the penetration mission, $\left(\dot{\sigma}_{intlos}, \dot{\theta}_{intlos}\right)$ is related to the overloads of the UAV and the interceptor, and the greater the value, the higher the costs of the interceptor at the interception terminal phase, and the UAV will break through more easily if the control overload of the interceptor reaches saturation.

## 5. DRL Penetration Guidance Law

### 5.1. SAC Training Model

Standard DRL maximizes the sum of expected rewards $\sum_t \mathbb{E}_{(s_t,a_t)\sim\rho_\pi}[f_r(s_t,a_t)]$. For the problem of multi-dimensional continuous state inputs and a continuous action output, SAC networks are introduced to solve the MDP model.

Compared with other policy learning algorithms [28], SAC augments the standard RL objective with expected policy entropy using Equation (47).

$$J_\pi = \sum_t \gamma^t \mathbb{E}_{(s_t,a_t)\sim\rho_\pi}[f_r(s_t,a_t) + \tau\mathcal{H}(\pi(\cdot\,|s_t))] \tag{47}$$

The entropy term $\tau\mathcal{H}(\pi(\cdot\,|s_t))$ is shown in Equation (48), which represents the stochastic feature of the strategy, balancing the exploration and learning of networks. The entropy parameter $\tau$ determines the relative importance of entropy against the immediate reward.

$$\begin{aligned}\mathcal{H}(\pi(\cdot\,|s_t)) &= -\int_{a\in A} \pi(a\,|s_t)\log\pi(a\,|s_t)da \\ &= \mathbb{E}_{a\sim\pi(\cdot\,|s_t)}[-\log\pi(a\,|s_t)]\end{aligned} \tag{48}$$

The optimal strategy of SAC is shown in the Equation (49), aiming at maximizing the cumulative reward and policy entropy.

$$\pi^*_{MaxEnt} = \underset{\pi}{\arg\max} \sum_t \gamma^t \mathbb{E}_{(s_t, a_t) \sim \rho_\pi} [f_r(s_t, a_t) + \tau \mathcal{H}(\pi(\cdot | s_t))] \tag{49}$$

The framework of SAC networks is shown in Figure 5, consisting of an Actor network and a Critic network. The Actor network generates the action, and the environment returns the reward and the next state. All of the ballistics data are stored in the experience pool, including the state, action, reward, and next state.

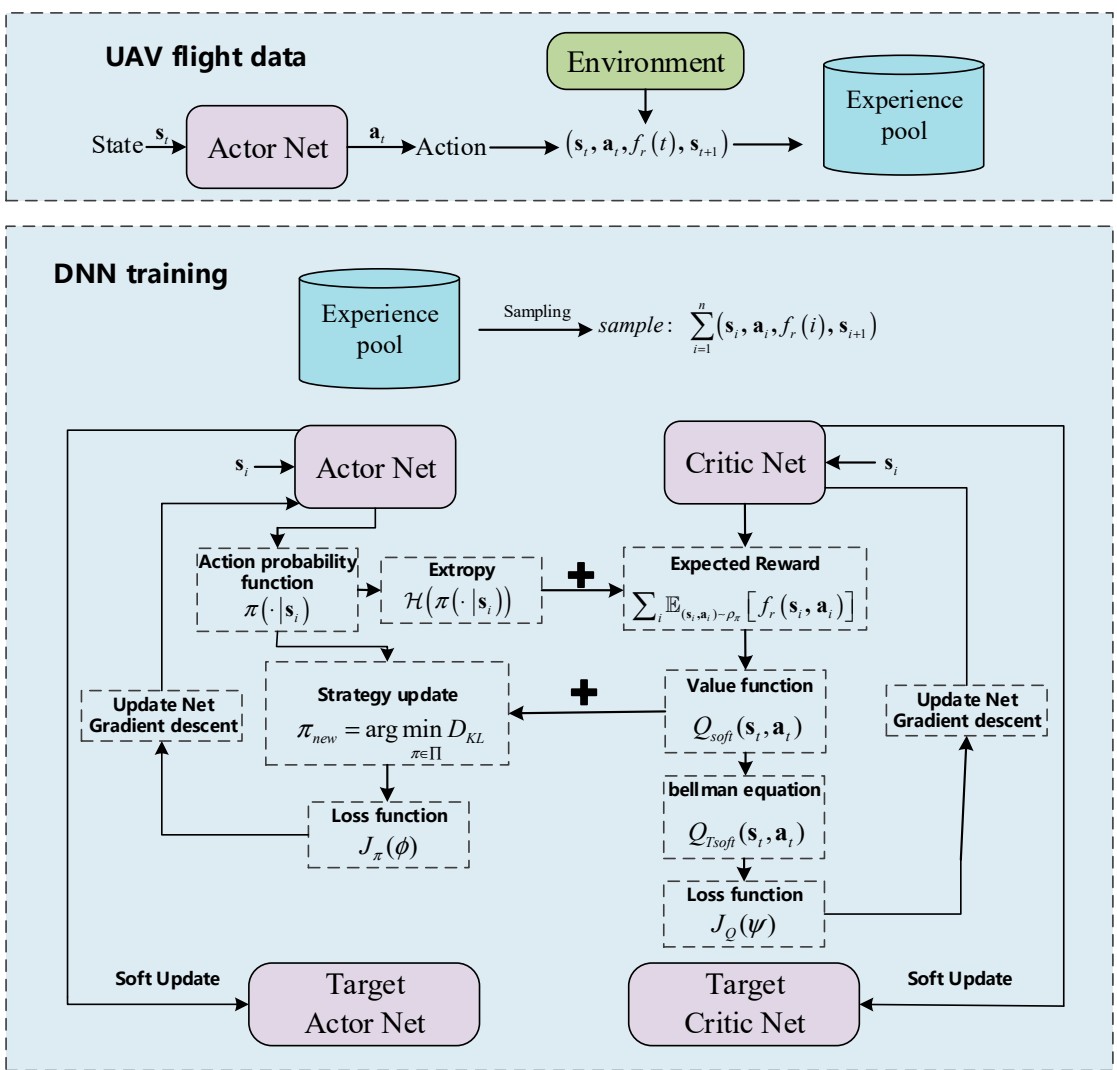

**Figure 5.** Updating principle of SAC.

The Critic network is used to judge the found strategies, which impartially guides the strategy of the Actor network. At the beginning, the Actor network and Critic network are given random parameters. The Actor network struggles to generate the optimal strategy, and Critic network struggles to scientifically judge the strategy of the Actor network. The parameters of the networks need to be updated based on continuously generating data and sampling ballistics data.

For updating the Critic network, it outputs the expected reward $\underset{a \sim \pi(\cdot | s_t)}{\mathbb{E}}$ based on samples, and the Actor network outputs the action probability, which is depicted by the

entropy term $\mathcal{H}(\pi(\,\cdot\,|s_t))$. Combining $\underset{a \sim \pi(\,\cdot\,|s_t)}{\mathbb{E}}$ with $\mathcal{H}(\pi(\,\cdot\,|s_t))$, the value function is conducted and shown in Equation (50)

$$Q_{soft}(\mathbf{s}_t, \mathbf{a}_t) = \underset{(\mathbf{s}_t, \mathbf{a}_t) \sim \rho_\pi}{\mathbb{E}} \left[ \sum_{t=0}^{\infty} \gamma^t r(\mathbf{s}_t, \mathbf{a}_t) + \tau \sum_{t=1}^{\infty} \gamma^t \mathcal{H}(\pi(\cdot|\mathbf{s}_t)) \right] \tag{50}$$

We further obtain the Bellman equation, as shown in Equation (51):

$$Q_{soft}(\mathbf{s}_t, \mathbf{a}_t) = \underset{\substack{\mathbf{s}_{t+1} \sim \rho(\mathbf{s}_{t+1}|\mathbf{s}_t, \mathbf{a}_t) \\ \mathbf{a}_{t+1} \sim \pi}}{\mathbb{E}} \left[ r(\mathbf{s}_t, \mathbf{a}_t) + \gamma \left( Q_{soft}(\mathbf{s}_{t+1}, \mathbf{a}_{t+1}) + \tau H(\pi(\cdot|\mathbf{s}_{t+1})) \right) \right] \tag{51}$$

Given by Equation (52), the loss function of the Critic network is acquired as follows:

$$J_Q(\psi) = \underset{\substack{(\mathbf{s}_t, \mathbf{a}_t, \mathbf{s}_{t+1}) \sim D \\ \mathbf{a}_{t+1} \sim \pi}}{\mathbb{E}} \left[ \frac{1}{2} \left( Q_{soft}(s_t, a_t) - \left( r(\mathbf{s}_t, \mathbf{a}_t) + \gamma \left( Q_{soft}(\mathbf{s}_{t+1}, \mathbf{a}_{t+1}) - \tau \log(\pi(\mathbf{a}_{t+1}|\mathbf{s}_{t+1})) \right) \right) \right)^2 \right] \tag{52}$$

For updating the Actor network, the updating strategy is shown in Equation (53)

$$\pi_{new} = \underset{\pi \in \Pi}{\arg\min} D_{KL} \left( \pi(\cdot|\mathbf{s}_t) \left\| \frac{\exp\left(\frac{1}{\tau} Q_{soft}^{\pi_{old}}(\mathbf{s}_t, \cdot)\right)}{Z_{soft}^{\pi_{old}}(\mathbf{s}_t)} \right) \right. \tag{53}$$

where $\Pi$ represents the set of strategies, and $Z$ is the partition function, used to normalize the distribution. $D_{KL}$ is the Kullback–Leibler (KL) divergence [29].

Combining the re-parameterization technique with Equation (54), the loss function of the Actor network is obtained as follows:

$$J_\pi(\phi) = \underset{\mathbf{s}_t \sim D, \varepsilon_t \sim N}{\mathbb{E}} [\tau \log \pi(f(\varepsilon_t; \mathbf{s}_t)|\mathbf{s}_t) - Q_{soft}(\mathbf{s}_t, f(\varepsilon_t; \mathbf{s}_t))] \tag{54}$$

in which $\mathbf{a}_t = f(\varepsilon_t; \mathbf{s}_t)$, and $\varepsilon_t$ is the input noise, obeying the distribution $N$.

The method of stochastic gradient descent is introduced to minimize the loss function of networks. The optimal parameters of the Actor–Critic networks are obtained by repeating the updating process and passing the parameters to the target networks via soft updating.

*5.2. Meta SAC Optimization Algorithm*

The learning algorithm in DRL relies on a lot of interaction between the agent and the environment and high training costs. Once the environment changes, the original strategy is no longer applicable and needs to be learned from scratch. The penetration guidance problem under the stable flight environment can be solved using SAC networks. For a changeable flight environment, such as where the initial position of the interceptor changes greatly or the interceptor guidance law deviates greatly from the preset value, the strategy solved via traditional SAC struggles to adapt, which thus requires to restudy and redesign. This manuscript introduces meta-learning to optimize and improve SAC performance. The training goal of Meta SAC is to obtain initial SAC model parameters. When the UAV penetration mission is changed, through a few scenes of learning, the UAV can adapt to the new environment and complete the corresponding guidance penetration mission, without relearning model parameters. Meta SAC can achieve "learn while flying" for UAVs and strengthen the adaptability of UAVs.

The Meta SAC algorithm is shown in Algorithm 1, which is divided into a meta-training and meta-testing phase. The meta-training phase seeks to determine the optimal meta-learning parameters based on multi-experience missions. In the meta-testing phase, the trained meta parameters are interactively learned in the new mission environment to fine-tune the meta parameters.

---

**Algorithm 1** Meta SAC

---

1: Initialize the experience pool $\Omega$, Storage space $N$
2: **Meta training:**
3: **Inner loop**
4: **for** iteration $k$ **do**
5:     sample mission($k$) from $\mathcal{T} \sim p(\mathcal{T})$
6:     update actor policy $\Theta$ to $\Theta'$ using SAC based on mission($k$):
7:     $\Theta' \leftarrow SAC(\Theta, mission(k))$.
8: **Outer loop**
9: $\Theta = mmse\left(\sum\limits_{i=1}^{k} \Theta'_i\right)$
10: Generate $\mathcal{D}_1$ from $\Theta$ and estimate the reward of $\Theta$.
11: Add a hidden layer feature as a random noise.
12: $\Theta'_i = \Theta + \alpha_\Theta \nabla_\Theta E_{\alpha_t \sim \pi(\alpha_t|s_t;\Theta,z_i), z_i \sim \mathcal{N}(\mu_i,\sigma_i)} \left[\sum\limits_t R_t(s_t)\right]$
13: The meta-learning process of different missions is carried out through SGD.
14: **for** iteration *mission*($k$) **do**
15:     $\min\limits_{\Theta} \sum\limits_{\mathcal{T}_i \sim p(\mathcal{T})} \mathcal{L}_{\mathcal{T}_i}(f_{\Theta'_i}) = \sum\limits_{\mathcal{T}_i \sim p(\mathcal{T})} \mathcal{L}_{\mathcal{T}_i}(f_{\Theta - \alpha \nabla \Theta \mathcal{L}_{\mathcal{T}_i}(f_\Theta)})$
16:     $\Theta = \Theta - \beta \nabla \Theta \sum\limits_{\mathcal{T}_i \sim p(\mathcal{T})} \mathcal{L}_{\mathcal{T}_i}(f_{\Theta'_i})$
17: **Meta testing**
18: Initialize the experience pool $\Omega$, Storage space $N$.
19: Load meta training network parameters $\Theta$.
20: Set training parameters.
21: **for** iteration $i$ **do**
22:     sample mission from $\mathcal{T} \sim p(\mathcal{T})$
23:     $\Theta' = SAC(\Theta, mission(k))$
**End for**

---

The basic assumption of Meta SAC is that the experience mission for meta training and the new mission for meta testing obey the same mission distribution $p(\mathcal{T})$. Therefore, there are some common characteristics between different missions. In the DRL scenario, our goal is to learn a function $f_\theta$ with parameter $\theta$, which can minimize the loss function $\mathcal{L}_\mathcal{T}$ of a specific mission $\mathcal{T}$. In the meta DRL scenario, our goal is to learn a learning process $\theta' = \mu_\psi(\mathcal{D}_\mathcal{T}^{tr}, \theta)$, which can quickly adapt to the new mission $\mathcal{T}$ with a very small dataset $\mathcal{D}_\mathcal{T}^{tr}$. Meta SAC can be summarized as optimizing the parameters $\theta$ and $\psi$ in the learning process, and the optimization equation is shown in Equation (55).

$$\min\limits_{\theta,\psi} E_{\mathcal{T} \sim P(\mathcal{T})}\left[\mathcal{L}(\mathcal{D}_\mathcal{T}^{test}, \theta')\right] \quad \text{s.t. } \theta' = \mu_\psi(\mathcal{D}_\mathcal{T}^{tr}, \theta) \tag{55}$$

where $\mathcal{D}_\mathcal{T}^{test}$ and $\mathcal{D}_\mathcal{T}^{tr}$, respectively, represent training and testing missions sampled from $p(\mathcal{T})$, and $\mathcal{L}(\mathcal{D}_\mathcal{T}^{test}, \theta')$ represents the testing loss function. In the meta-training phase, parameters are optimized via the inner loop and outer loop.

In the inner loop, Meta SAC updates the model parameters with a small amount of randomly selected data for the specific mission $\mathcal{T}$ as the training data, reducing the loss of the model in mission $\mathcal{T}$. In this part, the updating of model parameters is the same as in the original SAC algorithm, and the agent learns several scenes from randomly selected missions.

The minimum mean square error of strategy parameters $\theta$ corresponding to different missions in the inner loop phase is solved to obtain the initial strategy parameters $\theta_{ini}$ of the outer loop. In this manuscript, a hidden layer feature is added to the input part of strategy $\theta_{ini}$ as a random noise. The random noise is sampled again in each episode, in order to provide a more continuous random exploration in time, which is helpful for agent to adjust their overall strategy exploration according to the current mission MDP. The goal of meta-learning is to enable the agent to learn how to quickly adapt to new missions by



simultaneously updating a small amount of gradient of strategy parameters and hidden layer features. Therefore, the $\theta$ of $\theta' = \mu_\psi(\mathcal{D}_\mathcal{T}^{tr}, \theta)$ includes not only parameters of the neural network, but also the distribution parameters of hidden variables of each mission, namely the mean and variance of the Gaussian distribution, as shown in Equation (56).

$$
\begin{aligned}
\mu_i' &= \mu_i + \alpha_\mu \nabla_{\mu_i} E_{\alpha_t \sim \pi(\alpha_t|s_t;\theta,z_i), z_i \sim \mathcal{N}(\mu_i,\sigma_i)} \left[ \sum_t R_t(s_t) \right] \\
\sigma_i' &= \sigma_i + \alpha_\sigma \nabla_{\sigma_i} E_{\alpha_t \sim \pi(\alpha_t|s_t;\theta,z_i), z_i \sim \mathcal{N}(\mu_i,\sigma_i)} \left[ \sum_t R_t(s_t) \right] \\
\theta_i' &= \theta + \alpha_\theta \nabla_\theta E_{\alpha_t \sim \pi(\alpha_t|s_t;\theta,z_i), z_i \sim \mathcal{N}(\mu_i,\sigma_i)} \left[ \sum_t R_t(s_t) \right]
\end{aligned}
\tag{56}
$$

The model is represented by a parameterized function $f_\theta$ with parameter $\theta$, and when it is transferred to a new mission $\mathcal{T}$, model parameter $\theta$ is updated to $\theta'$ through gradient rise, as shown in Equation (57).

$$
\theta_i' = \theta - \alpha \nabla_\theta \mathcal{L}_{\mathcal{T}_i}(f_\theta)
\tag{57}
$$

We update step $\alpha$ is a fixed super parameter. Model parameter $\theta$ is updated to maximize the performance $f_{\theta_i'}$ of different missions, as shown in Equation (58).

$$
\min_\theta \sum_{\mathcal{T}_i \sim p(\mathcal{T})} \mathcal{L}_{\mathcal{T}_i}\left(f_{\theta_i'}\right) = \sum_{\mathcal{T}_i \sim p(\mathcal{T})} \mathcal{L}_{\mathcal{T}_i}(f_{\theta-\alpha\nabla\theta\mathcal{L}_{\mathcal{T}_i}(f_\theta)})
\tag{58}
$$

The meta-learning process of different missions is carried out through SGD, and the principle of $\theta$ is as follows:

$$
\theta = \theta - \beta \nabla_\theta \sum_{\mathcal{T}_i \sim p(\mathcal{T})} \mathcal{L}_{\mathcal{T}_i}\left(f_{\theta_i'}\right)
\tag{59}
$$

where $\beta$ is the meta update step.

In the meta-testing phase, a small amount of experience in new missions is used to quickly learn strategies for solving new missions. A new mission may involve completing a new mission goal or achieving the same mission goal in a new environment. The updating process of the model in this phase is the same as the cycle part in the meta-training phase, and by calculating the loss function with the data collected in the new mission and adjusting the model through back propagation, the new mission is adapted by the agent.

## 6. Simulation Analysis

In this section, the manuscript analyzes and verifies the escape guidance strategy based on Meta SAC. SAC is used to solve a specific escape guidance mission. We conduct comprehensive experiments to verify whether the UAV can complete the guidance escape mission under satisfying terminal and process constraints. Once the UAV guidance escape mission changes, the original strategy based on SAC cannot be easily adapted to the changed mission and thus needs to be relearned and retrained. The manuscript proposes an optimization method via meta-learning that improves the learning ability of UAVs during the training process. This section focuses on verifying the validation of Meta SAC and demonstrating its performance in various new missions. In addition, the maneuvering overload commands under different pursuit evading distances are analyzed in order to explore the influence of different maneuvering timings and distances on the escape results. We use CAV-H to verify the escape guidance performance. The initial conditions, terminal altitude, and Meta SAC training parameters are given in Table 1.

**Table 1.** Simulation and Meta SAC training conditions.

| Scheme | | Meta SAC Training Parameters | |
|---|---|---|---|
| UAV initial velocity | 4000 m/s | Learning episodes | 1000 |
| Initial velocity Inclination | 0° | Guidance period | 0.1 s |
| Initial velocity azimuth | 0° | Data sampling interval | 30 Km |
| Initial position | (3° E, 1° N) | Discount factor | $\gamma = 0.99$ |
| Initial altitude | 45 Km | Soft update tau | 0.001 |
| Terminal altitude | 40 Km | Learning rate | 0.005 |
| Target position | (0° E, 0° N) | Sampling size for each train | 128 |
| Interceptor Initial velocity | 1500 m/s | Net layers | 2 |
| Initial velocity Inclination | Longitudinal LOS angle | Net nodes | 256 |
| Initial velocity azimuth | Lateral LOS angle | Capacity of experience pool | 20,000 |

*6.1. Validity Verification on SAC*

In order to verify the effectiveness of SAC, three different pursuit evading scenarios are constructed, and the terminal reward value, miss distance, and terminal position deviation are analyzed. As shown in Figure 6a, the terminal reward value is poor in the initial phase of training, which demonstrates that the optimal strategy is not found. After 500 episodes, the terminal reward value increases gradually, indicating that a better strategy has been explored and converged. In the last 100 episodes, the optimal strategy is trained and learned, while the network parameters are adjusted to the optimal strategy. As can be seen from Figure 6b, the miss distance is relatively divergent in the first 150 episodes of training, indicating that the Action network in SAC constantly explores new strategies, and the Critic network also learns scientific evaluation criteria. After 500 training episodes, the network gradually learns and trains in the direction of optimal solution. The miss distance at the encounter moment converges to about 20 m. As shown in Figure 6c, the terminal position of the UAV has a large deviation in the early training phase, which is attributed to the exploration of the escape strategy by the network. In the later training phase, the position deviation is less affected by exploration. These pursuit evading scenarios tested in the manuscript can achieve convergence, and the final convergence values are all within 1 m.

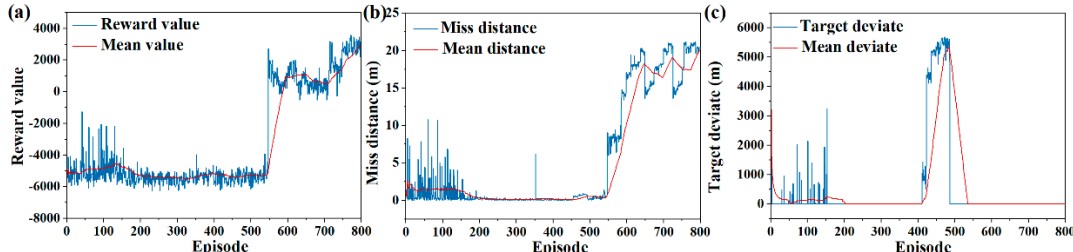

**Figure 6.** Train results of SAC of the short-range scenario. (**a**) Reward value, (**b**) miss distance, (**c**) target deviation.

In order to verify whether the SAC algorithm can solve the escape guidance strategy that meets the mission requirements in different pursuit and evasion scenarios, the pursuing and evading distance is changed, and the training results are shown in Figure 7. In the medium-range scenario, the miss distance converges to about 2 m, and the terminal deviation converges to about 1 m.

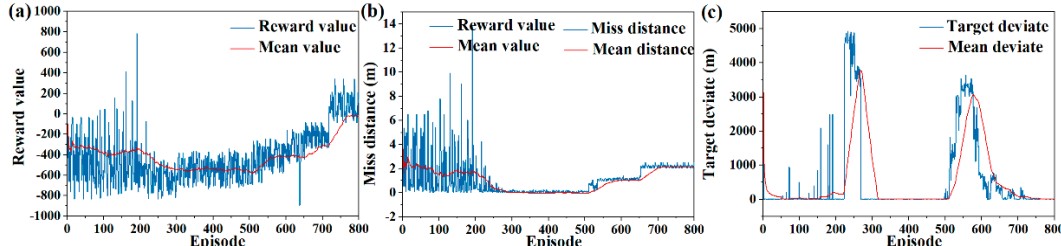

**Figure 7.** Train results of SAC of the medium-range scenario. (**a**) reward value, (**b**) miss distance, (**c**) target deviation.

As shown in Figure 8, in long-range attack and defense scenarios, the miss distance converges to about 5 m, and the terminal deviation converges to about 1 m.

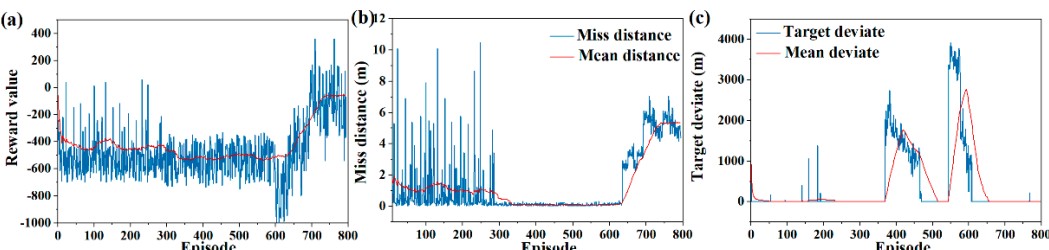

**Figure 8.** Train results of SAC of the long-range scenario. (**a**) reward value, (**b**) miss distance, (**c**) target deviation.

Based on the above simulation analysis, SAC is a feasible method to solve the UAV guidance escape strategy. After limited episodes of learning and training, network parameters are converged, which is used to test the flight mission.

*6.2. Validity Verification on Meta SAC*

When the mission of the UAV changes, the original SAC parameters cannot meet the requirements of the new mission, and thus the parameters need to be retrained and relearned. The SAC proposed in the manuscript is improved via meta-learning. Strong adaptive network parameters are found using learning and training, and when the pursuit evading environment changes, the network parameters are fine-tuned to adapt to the new environment immediately.

Meta SAC is divided into a meta-training phase and a meta-testing phase, and initialization parameters for the SAC network are trained in the meta-training phase, which is fine-tuned by interacting with the new environment in the meta-testing phase. By changing the initial interceptor position, three different pursuit evading scenarios are constructed, which represent a short, medium, and long distance.

The training results of Meta SAC and SAC are compared, and terminal reward values are presented in Figure 9a. Meta SAC is an effective method to speed up the training process, and after 100 episodes, a better strategy is learned by the network and converged gradually. In contrast, the SAC network needs 500 episodes to find the optimal solution. Miss distance is shown in Figure 9b. The better strategy is quickly learned by Meta SAC, which is more effective than the SAC method. Figure 9c shows the terminal deviation between the UAV and the target.

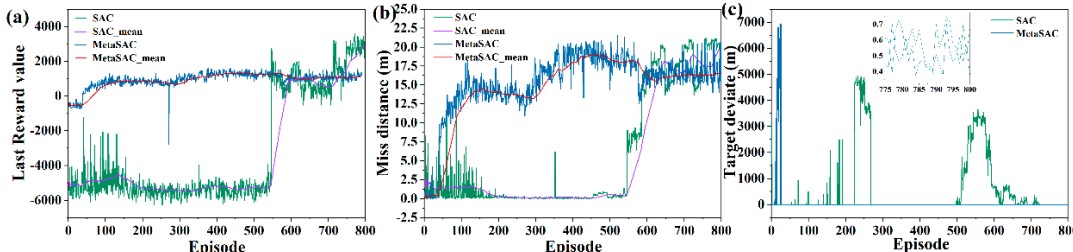

**Figure 9.** Meta SAC training results of the short-range scenario. (**a**) Reward value, (**b**) miss distance, (**c**) target deviation.

To explore the optimal solution as much as possible, some strategies with large terminal position deviations appear in the training process. As shown in Figure 10b,c, in medium-range attack and defense scenarios, the miss distance converges to about 8 m based on Meta SAC, and the terminal deviation converges to about 1 m.

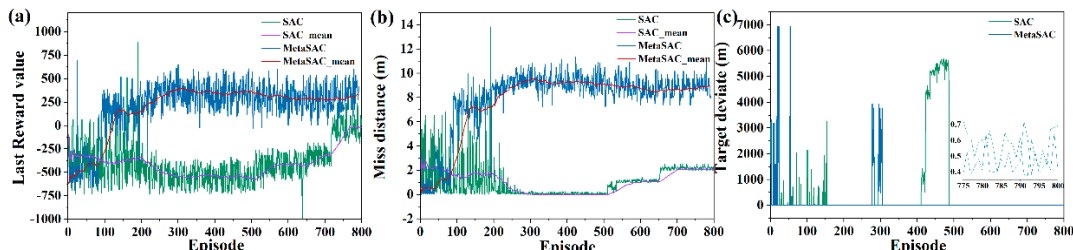

**Figure 10.** Meta SAC training results of the medium-range scenario. (**a**) Reward value, (**b**) miss distance, (**c**) target deviation.

As shown in Figure 11b,c, in long-range attack and defense scenarios, the miss distance converges to about 10 m based on Meta SAC, and the terminal deviation converges to about 1 m.

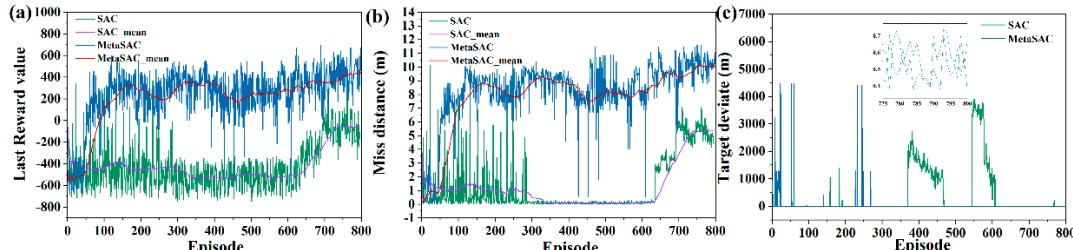

**Figure 11.** Meta SAC training results of the long-range scenario. (**a**) Reward value, (**b**) miss distance, (**c**) target deviation.

According to the theoretical analysis, in the training process, new missions corresponding to the same distribution are used to execute micro-testing using Meta SAC, resulting in more gradient-descending directions of the optimal solution being learned by the network. Combined with the theory analysis and training results, the manuscript demonstrates that meta-learning is a feasible method to accelerate convergence and improve the efficiency of training.

In the previous analysis, when the pursuit evading scenario is changed, network parameters obtained in the meta-training phase are fine-tuned through a few interactions. The manuscript verifies meta testing performance by changing the initial interceptor position, and results compared with the SAC method are shown in Table 2. Based on the network parameters of the meta-training phase, the strategic solutions for the escape guidance missions are found through training within 10 episodes. On the contrary, network parameters

based on SAC need more interaction to find solutions, and the episode of interactions comprises more than 50 episodes. According to the above simulation, the adaptability of Meta SAC is much greater than SAC, and once the escape mission changes, through very few episodes of learning, the new mission is completed by the UAV without re-learning and designing the strategy. The method provides the possibility of realizing UAV learning while flying.

**Table 2.** Results compared with the SAC method.

| Interceptor Initial Position (Km) | Interaction Episodes | | Miss Distance (m) | | Terminal Deviate (m) | |
|---|---|---|---|---|---|---|
| | SAC | Meta SAC | SAC | Meta SAC | SAC | Meta SAC |
| (0, 30, 0) | 74 | 1 | 3.78 | 3.29 | 0.56 | 0.61 |
| (2, 30, 6) | 75 | 4 | 2.80 | 2.72 | 0.68 | 0.72 |
| (4, 30, 12) | 59 | 8 | 6.93 | 3.75 | 0.69 | 0.58 |
| (6, 30, 18) | 59 | 1 | 2.71 | 6.82 | 0.68 | 0.72 |
| (8, 30, 24) | 26 | 2 | 3.16 | 3.70 | 0.47 | 0.50 |
| (10, 30, 30) | 58 | 3 | 3.50 | 2.37 | 0.61 | 0.64 |
| (12, 30, 36) | 67 | 1 | 2.86 | 2.21 | 0.68 | 0.45 |
| (14, 30, 42) | 56 | 8 | 2.18 | 2.89 | 0.55 | 0.61 |
| (16, 30, 48) | 69 | 1 | 2.73 | 2.23 | 0.61 | 0.72 |
| (18, 30, 54) | 106 | 1 | 2.45 | 3.71 | 0.56 | 0.63 |
| (20, 30, 60) | 94 | 1 | 2.7 | 2.35 | 0.49 | 0.54 |
| (22, 30, 66) | 59 | 1 | 2.23 | 2.51 | 0.73 | 0.71 |
| (24, 30, 72) | 62 | 1 | 2.11 | 3.47 | 0.48 | 0.67 |
| (26, 30, 78) | 63 | 1 | 2.04 | 4.5 | 0.48 | 0.57 |
| (28, 30, 84) | 63 | 4 | 2.64 | 5.12 | 0.47 | 0.40 |
| (30, 30, 90) | 63 | 9 | 2.95 | 6.05 | 0.68 | 0.47 |

*6.3. Strategy Analysis Based on Meta SAC*

This section tests the network parameters based on Meta SAC, and analyzes the escape strategy and flight state under different pursuit evading distances. As shown in Figure 12a, for the pursuit evading scene over a short distance, the longitudinal maneuvering overload is larger in the first half of the phase of escape, resulting in the velocity slope angle decreasing gradually. In the second half of the phase of escape, if the strategy is executed under the original maneuvering overload, the terminal altitude constraint cannot be satisfied, and therefore, the overload gradually decreases, and the velocity slope angle is slowly reduced. As shown in Figure 12b, at the beginning of escape, the lateral maneuvering overload is positive, and the velocity azimuth angle constantly increases. With the distance between the UAV and the interceptor reducing, the overload increases gradually in the opposite direction, and the velocity azimuth angle decreases. On the one hand, this can confuse the strategy of the interceptor, while on the other hand, the guidance course is corrected.

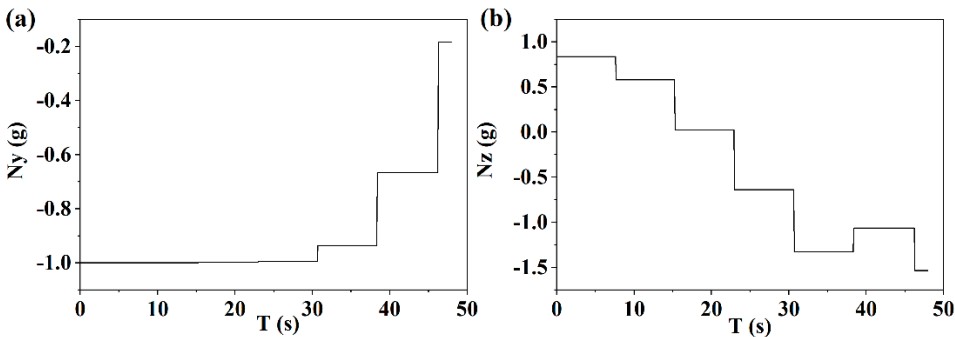

**Figure 12.** The maneuvering overload. (**a**) Longitudinal direction under a short distance, (**b**) lateral direction under a short distance.

As shown in Figure 13a, compared with the pursuit evading scene of a short distance, the medium-distance escape process takes longer, the pursuing time left to the interceptor is longer, and the UAV flies in the direction of increasing the velocity slope angle. The timing of the maximum escape overload corresponding to the medium distance is also different. As shown in Figure 13b, in the first half of the phase of escape, lateral maneuvering overload corresponding to a medium distance is larger than that in a short distance, and in the second half of the phase of escape, the corresponding reverse maneuvering overload is smaller, resulting in the UAV being able to use the longer escape time to slowly correct the course.

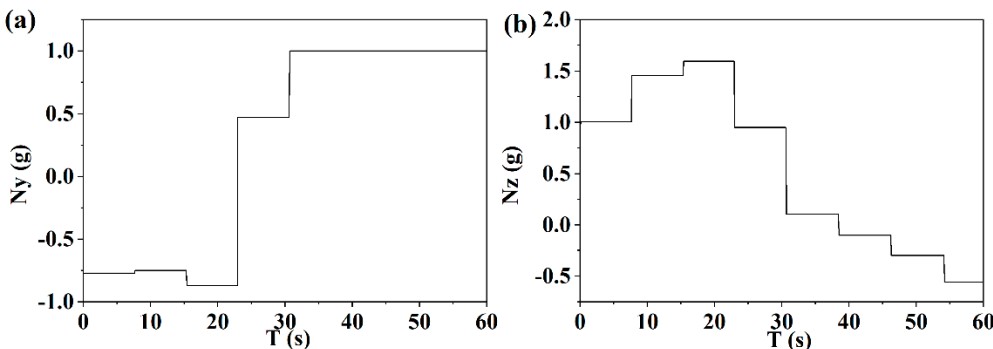

**Figure 13.** The maneuvering overload. (**a**) Longitudinal direction under a medium distance, (**b**) lateral direction under a medium distance.

As shown in Figure 14, under a long pursuit distance, the overload change of the UAV maneuver is similar to that for a medium range, and the escape timing is basically the same as the escape strategy.

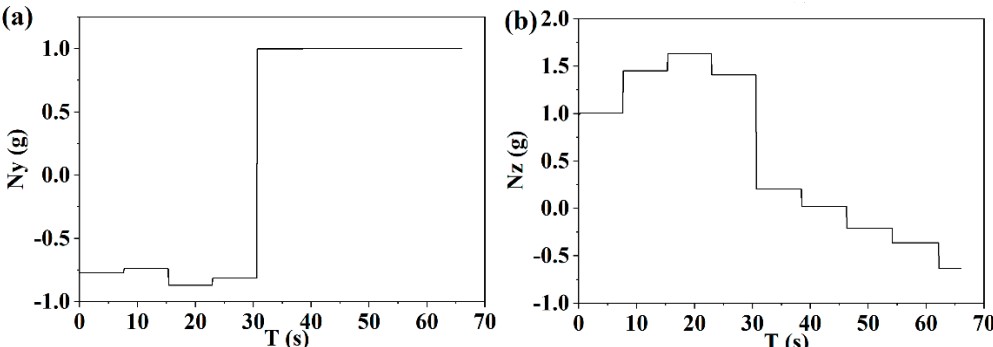

**Figure 14.** The maneuvering overload. (**a**) Longitudinal direction under a long distance, (**b**) lateral direction under a long distance.

According to the above analysis, the escape guidance strategy via Meta SAC can be used as a tactical escape strategy, and the timing of escape and the maneuvering overload are adjusted in a timely manner under different pursuit evading distances. On the one hand, the overload corresponding to this strategy can confuse the interceptor and cause some interference, and on the other hand, it can take into account the guidance mission, correcting the course deviation caused by escape.

Figure 15a shows the flight trajectory of the interceptor against UAV at the North East Down (NED) coordinate (10 km, 30 km, 30 km), the trajectory point at the encountering moment is shown in Figure 15b, and the miss distance is 19 m in this pursuit evading scene. To verify the scientific and applicability of Meta SAC, the initial position of interceptor is changed. Flight trajectories are shown in Figure 15c,e, and trajectory points at the encountering moment are shown in Figure 15d,f. The miss distances in these two pursuit evading scenarios are 3 m and 6 m, respectively. Based on the CAV-H structure, the miss

distance between the UAV and the interceptor is greater than 2 m at the encountering moment, which means that the escape mission is achieved.

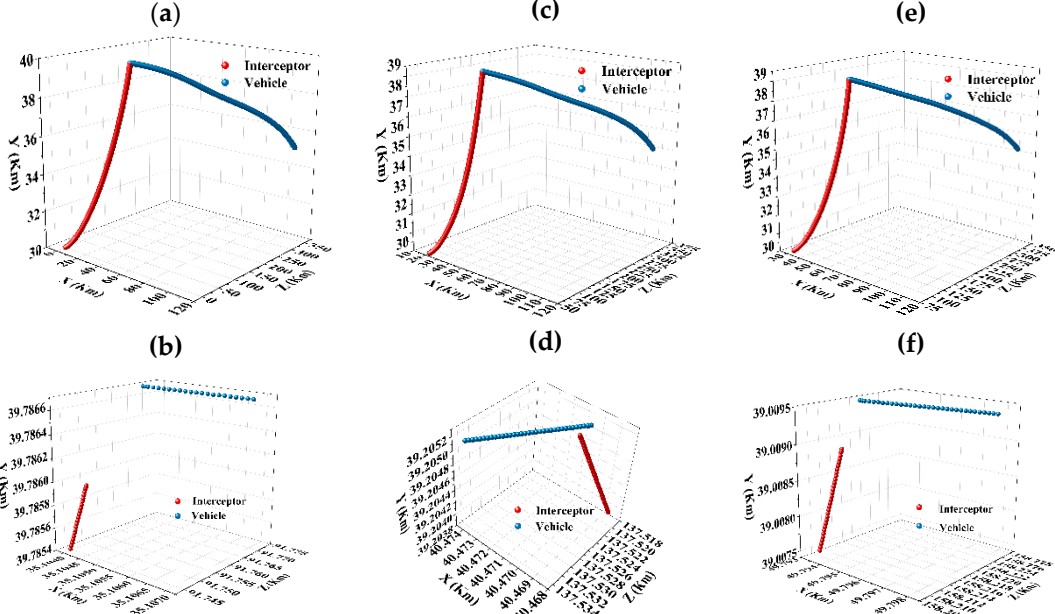

**Figure 15.** The ballistic flight diagrams of the whole course under different pursuit evading distances. (**b**,**d**,**f**) represent trajectories at the encountering time under different pursuit evading distances.

Based on the principles of Meta SAC and optimal guidance, flight states are shown in Figure 16. Longitude, latitude, and altitude during the flight of the UAV are shown in Figure 16a,b, under different pursuit evading scenarios, and the terminal position and altitude constraints are met. There is larger amplitude modification in the velocity slope and azimuth angle, which is attributed to the escape strategy via lateral and longitudinal maneuvering, as shown in Figure 16c,d. The total change in velocity slope and azimuth angle is within two degrees, which meets the flight process constraints. Through the analysis of flight states, this escape strategy is an effective measure for guidance and escape with high accuracy.

Flight process deviation mainly includes aerodynamic calculation deviation and output overload deviation. For the aerodynamic deviation, this manuscript uses the interpolation method to calculate it based on the flight Mach number and angle of attack, which may have some deviation. Therefore, when calculating the aerodynamic coefficient, random noise with an amplitude of 0.1 is added to verify whether the UAV can complete the guidance mission. As shown in Figure 17a, aerodynamic deviation noise causes certain disturbances to the angle of attack during the flight. At the 10th second and end of the flight, the maximum deviation of the angle of attack is 2°. However, overall, the impact of aerodynamic deviation on the entire flight is relatively small, and the change in angle of attack is still within the safe range of the UAV. As shown in Figure 17b, due to the constraints of UAV game confrontation and guidance missions, the bank angle during the entire flight process changes significantly, and aerodynamic deviation noise has a small impact on the bank angle. After increasing the aerodynamic deviation noise, the miss distance between the UAV and the interceptor at the time of encounter is 8.908 m, and the terminal position deviation is 0.52 m. Therefore, under the influence of aerodynamic deviation, the UAV can still complete the escape guidance mission.

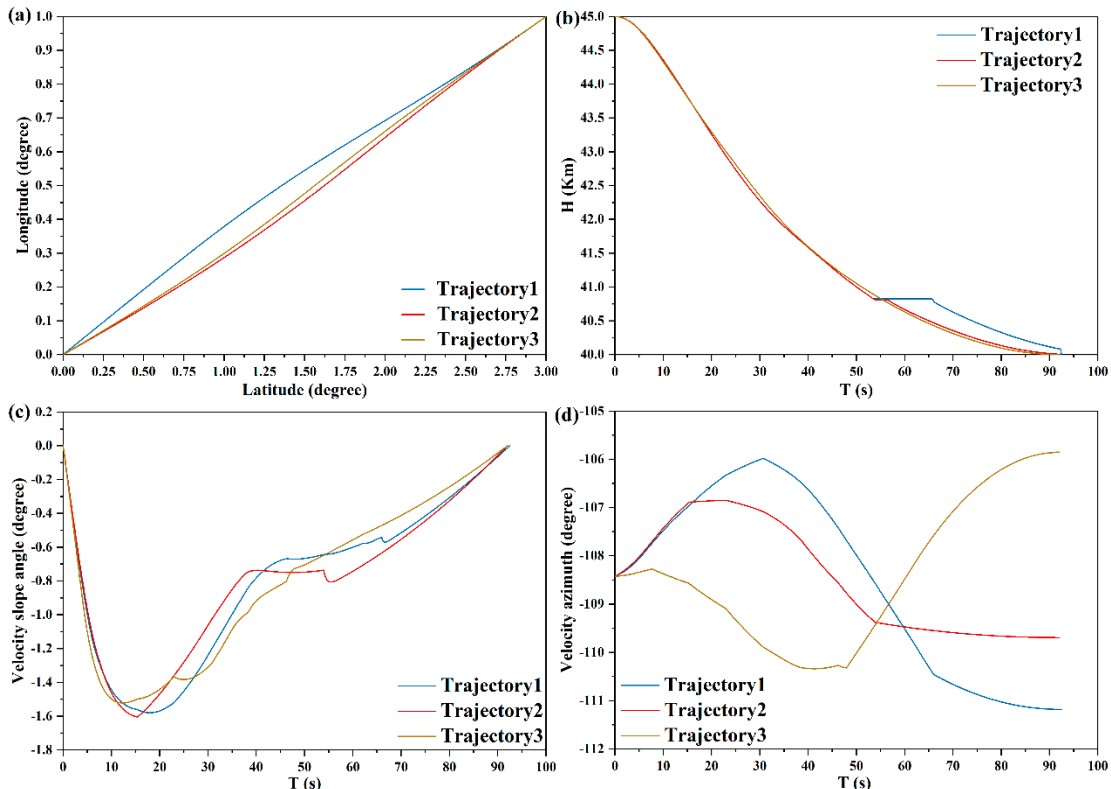

**Figure 16.** Flight states of the UAV. (**a**) The longitude and latitude, (**b**) the height, (**c**) the velocity slope angle, (**d**) the velocity azimuth angle.

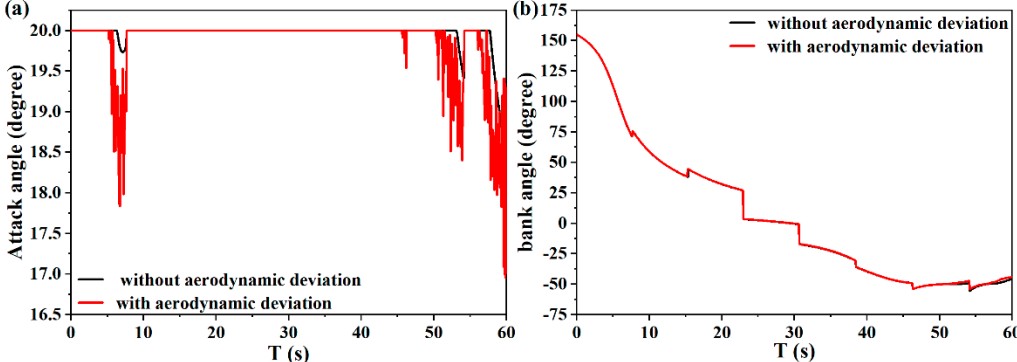

**Figure 17.** (**a**) The attack angle, (**b**) the bank angle.

For the output overload deviation, the total overload is composed of the guiding overload derived from the optimal guidance law and the maneuvering overload output derived from the neural network. Random maneuvering overload with an amplitude of 0.1 is added to verify whether the UAV can complete the maneuver guidance mission. As shown in Figure 18, random overloads are added in the longitudinal and lateral directions, respectively. Through simulation testing, the miss distance between the UAV and the interceptor at the encounter point is 10.51 m, and the terminal deviation of the UAV is 0.6 m. Under this deviation, the UAV can still achieve high-precision guidance and efficient penetration.

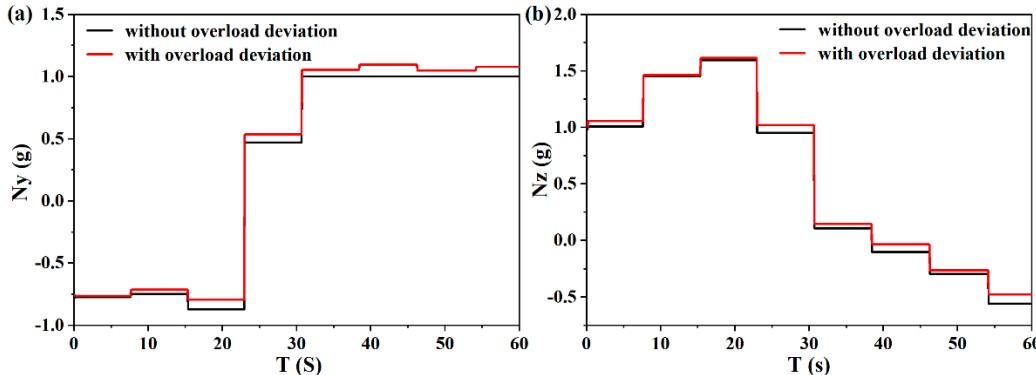

**Figure 18.** (**a**) The maneuvering overload in the longitudinal direction, (**b**) the maneuvering overload in the lateral direction.

## 7. Conclusions

This manuscript proposes an escape guidance strategy satisfying multiple terminal constraints via SAC. The action space is designed under the UAV process constraint. Considering the fact that real-time interceptor information is hard to obtain in the actual escape process, the state space considering the target and the heading angle is designed. The reward function is an important index function that is used to guide and evaluate the training results. Based on the pursuit evading model, the terminal LOS angle rates in the lateral and longitudinal directions are derived to describe the deviation and miss distance. In order to improve the adaptability of the escape guidance strategy, the manuscript improves SAC via meta-learning and compares meta SAC with SAC. The strong adaptive escape strategy based on Meta SAC is analyzed. In view of the above theoretical numerical analysis, we have obtained the following conclusions:

(1) The escape guidance strategy based on SAC is a feasible tactical escape strategy that can achieve high-precision guidance while meeting the escape requirements.
(2) Meta SAC can significantly improve the adaptability of the escape strategy. When the escape mission changes, it can fine-tune network parameters to adapt to the mission through a small number of training epochs. This method provides the possibility for the UAV to learn while flying.
(3) The strong adaptive escape strategy based on Meta SAC can adjust the escape timing and maneuver overload in real time according to the pursuit evading distance. On the one hand, the overload corresponding to this strategy can confuse the interceptor and cause some interference, while on the other hand, it can take into account the guidance mission, correcting the course deviation caused by escape.

**Author Contributions:** Conceptualization, S.Z. and J.Z.; methodology, S.Z.; software, S.Z; validation, S.Z. and J.Z.; formal analysis, S.Z; investigation, S.Z.; resources, X.L.; data curation, H.S.; writing—original draft preparation, H.S.; writing—review and editing, X.L.; visualization, S.Z.; supervision, W.B. All authors have read and agreed to the published version of the manuscript.

**Funding:** This research received no external funding.

**Data Availability Statement:** No new data were created or analyzed in this study. Data sharing is not applicable to this article.

**Conflicts of Interest:** The authors declare no conflict of interest.

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
