# Peer review of "A Multi-Constraint Guidance and Maneuvering Penetration Strategy via Meta Deep Reinforcement Learning"

_drones, doi:10.3390/drones7100626_

Round 1

Reviewer 1 Report

Typos should be carefully reviewed and corrected in the revised manuscript.

Author Response

Dear Reviewer: #1

Thank you very much for your enlightening suggestions on the revision of our manuscript, which are very comprehensive and targeted. We have made due modifications and corresponding explanations following your comments in the email, and we feel that the logic and integrity of the revised manuscript are clearer. The following will be a detailed explanation of the key problems you pointed out.

Reviewer#1, Concern # 1:
1. The references cited in this paper are incomplete, and there are several recent studies that have utilized meta-reinforcement learning for trajectory design in UAVs. The authors should make an effort to compare their work with these recent methods. Some notable examples include the works of Lu et al. (Lu, Z., Wang, X., & Gursoy, M. C. (2023, May). Trajectory Design for Unmanned Aerial UAVs via Meta Reinforcement Learning. In IEEE INFOCOM 2023-IEEE Conference on Computer Communications Workshops (INFOCOM WKSHPS) (pp. 1-6). IEEE), Hu et al. (Hu, Ye, et al. "Meta-reinforcement learning for trajectory design in wireless UAV networks." GLOBECOM 2020-2020 IEEE Global Communications Conference. IEEE, 2020), Yu et al. (Yu, Q., Luo, L., Liu, B., & Hu, S. (2023). Re-planning of Quadrotors Under Disturbance Based on Meta Reinforcement Learning. Journal of Intelligent & Robotic Systems, 107(1), 13), and Belkhale et al. (Belkhale, S., Li, R., Kahn, G., McAllister, R., Calandra, R., & Levine, S. (2021). Model-based meta-reinforcement learning for flight with suspended payloads. IEEE Robotics and Automation Letters, 6(2), 1471-1478). The authors should also mention the relevant research in the recent literature and give a brief overview of related research in the Introduction Section

Author response: Thank you for pointing out the insufficiency, and thank you very much for the papers listed by the reviewer, which are authoritative and scientific research achievements in the field of trajectory design in UAVs and meta-reinforcement learning, which are of great significance to improve and upgrade this manuscript.

We analyze the listed articles in the manuscript,

#1 Trajectory Design for Unmanned Aerial UAVs via Meta Reinforcement Learning

This paper mainly solved maximizing the total data collected and avoidance collisions during the guidance flight, and improved the adaptability of different tasks via meta RL. Through the simulation results and algorithm comparison, the proposed method has better performance.

#2 Meta-reinforcement learning for trajectory design in wireless UAV networks

The paper mainly studied a challenging trajectory design problem, optimized the DBS trajectory and considered the uncertainly and dynamic of terrestrial users ‘service request.

3# Re-planning of Quadrotors Under Disturbance Based on Meta Reinforcement Learning

MAML is introduced for solving raised POMDPS and enhancing adaptability to balance flight aggressiveness and safety.

#4 Model-based meta-reinforcement learning for flight with suspended payloads

For suspended payload transportation tasks, the paper proposed meta-learning approach to improve adaptability. The simulation demonstrate improvement in closed-loop performance compared to non-adaptive methods.

We have studied and researched recent relevant literature, and supplemented it in the introduction section.

Reviewer#1, Concern # 2:
The full name of "L/D" (lift-to-drag ratio) should be provided at its initial mention on page 4

Author response: Thank you for pointing out the insufficiency, we are sorry for the errors caused by carelessness. We have made revisions in the manuscript.

Reviewer#1, Concern # 3:
The authors should deliberate on whether to include a comparison with state-of-the-art research from recent literature, such as the aforementioned work of Lu et al. or Belkhale et al., in the Simulation analysis Section to validate the effectiveness of the proposed approach

Author response: Thank you very much for the reviewer's suggestions and careful review. After analyzing the relevant research papers, some articles that use meta learning algorithms mainly focus on UAV communication and optimizing paths. The innovation of this manuscript lies in the analysis of UAV guidance penetration problems, while considering the impact of maneuvering on the original pure guidance law when solving the strategy of maneuvering penetration. In section 6.1, we conducted simulation analysis on the training model using only the SAC algorithm, and the simulation results showed that this method can train strategy solutions that meet the requirements of UAV maneuvering penetration. In section 6.2, we compared the SAC algorithm with the improved algorithm based on meta learning. From the simulation results, we can obtain the adaptability of Meta SAC is much greater than SAC, once the escape mission changing, through very few episodes of learning, the new mission is completed by UAV without re-learning and designing strategy. The method provides possibility for realizing UAV learning while flying. In Section 6.3, we tested the model trained by Meta SAC, analyzed and summarized the penetration laws of the trained maneuver strategies. In addition, we also conducted simulation analysis on scenarios with added aerodynamic and overload deviations. Based on the analysis, the strong adaptive escape strategy based on meta SAC can adjust the escape timing and maneuvering overload in real time according to the pursuit evading distance.

Reviewer#1, Concern # 4:
The reference list should include the journal titles.

Author response: Thank you for pointing out the insufficiency, we have checked the format of the references and supplemented the titles of the journal.

Reviewer#1, Concern # 5:
Typos should be carefully reviewed and corrected in the revised manuscript.

For instance,(i) Abstract: The "which" in the third-to-last line of the abstract should be removed.

(ii) The grammar in the descriptions of the second and third points concerning the core contributions of this paper in the first section needs to be corrected.

(iii) The font size in the title of Fig. 2 is inconsistent.

(iv) Where multiple equations are presented, 'where' should be added before the parameter explanations following each equation

Author response: Thank you very much for pointing out the errors in the manuscript. We apologize for any grammar and detail errors that may have occurred in the manuscript. In view of these grammatical problems, we carefully examined the manuscript and corrected it. Once again, we thank the reviewers for their careful guidance.

Reviewer 2 Report

Minor corrections are needed.

Author Response

Dear Reviewer#2: Thank you for your suggestion of revision. According to your suggestion, we have made detailed modifications to the article, and I feel that the clarity of the article has been significantly improved.  Please see the attachment.
